## EQUITY, DIVERSITY AND INCLUSION

# Mental health in medical and biomedical doctoral students during the 2020 COVID-19 pandemic and racial protests

**Abstract** Concerns about the mental health of students, trainees and staff at universities and medical schools have been growing for many years. Recently, these have been exacerbated by the COVID-19 pandemic and a period of heightened reckoning and protests about systemic racism in the United States in 2020. To better understand the mental health of medical students and biomedical doctoral students at the University of North Carolina at Chapel Hill during this challenging period, we performed a cross-sectional study (n=957) using institutional annual survey data on measures of depression, anxiety, hazardous alcohol use, problems related to substance use, and suicidal ideation. These data were collected in 2019 and 2020, and were analyzed by type of training program, race/ethnicity, gender, sexual orientation, and survey year. Results indicated significant differences for rates of depression, anxiety, and suicidal ideation, with biomedical doctoral students showing greater incidence than medical students, and historically excluded students (e.g., people of color, women, LGBQ+ trainees) showing greater incidence compared to their peers. Of note, mental health remained poor for biomedical doctoral students in 2020 and declined for those belonging to historically excluded populations. The high rates of depression, anxiety, and suicidal ideation reported suggest that training environments need to be improved and support for mental health increased.

**ALLISON SCHAD\*[†], REBEKAH L LAYTON\*[†], DEBRA RAGLAND AND JEANETTE GOWEN COOK**

**\*For correspondence:**
allison_schad@med.unc.edu (AS);
rlayton@unc.edu (RLL)

[†]These authors contributed
equally to this work

**Competing interest:** The authors
declare that no competing
interests exist.

Reviewing Editor: Elsa Loissel,
eLife, United Kingdom

## Introduction

Graduate and professional programs in the biomedical and health sciences are stressful environments for junior researchers and clinicians. Indeed, the mental health of medical students declines during medical school and is worse than that of the general population by graduation (*Dyrbye et al., 2005*; *Dyrbye et al., 2006*; *Brazeau et al., 2014*). In fact, poor mental health in medical education is evidenced by a robust, decades-old body of literature, including a systematic review of mental health evidence-based research encompassing nearly 200 relevant studies to date (e.g., *Slavin et al., 2014*; *Rotenstein et al., 2016*; *Quek et al., 2019*).

In comparison, evidence-based research on mental health of doctoral students has only begun to emerge over the past five years. Recent reports indicate elevated rates of anxiety and depression among graduate students in all disciplines (*UC Berkeley Graduate Assembly, 2014*; *Evans et al., 2018*; *Nagy et al., 2019*; *Levecque et al., 2017*). For example, a multi-campus study of undergraduate and graduate students found that 17.3% scored positive for depression, 9.8% scored positive for anxiety, and 6.3% reported suicidal ideation (*Eisenberg et al., 2013*). One study found that graduate students in the Faculty of Medicine at the University of Calgary have higher anxiety and depression than undergraduates (*Toews et al., 1993*), and a survey of graduate students at a university in the United States showed that approximately half of respondents reported emotional or stress-related problems (*Hyun et al., 2006*). In an international sample of Masters' and PhD students, 39% of respondents indicated moderate to severe depression, and 41% moderate to severe anxiety scores (*Evans*

*et al., 2018*). In comparison, the Centers for Disease Control and Prevention (CDC) estimate that approximately 16% of adults in the United States experience anxiety, and approximately 5% depression. Moreover, approximately 1 in 6 Americans will experience depression in their lifetime, and more women than men experience anxiety (*CDC, 2021*; *Terlizzi and Villarroel, 2020*). However, while these general population incidences can be informative as a benchmark, these rates are not truly comparable to the student population due to multiple potential confounding factors (e.g., age group, socioeconomic status, education levels of graduate level students, etc.).

Less is known about the mental health of biomedical doctoral students specifically, as disparate disciplines are often combined when graduate students are included in student mental health studies. Still, these students train in environments which may feature long work hours, pressures to produce, influential and sometimes unsupportive relationships with advisors, as well as financial concerns, uncertainty about future employment, and non-transparent university processes which may all negatively impact doctoral student mental health (*Hazell et al., 2020*; *Mackie and Bates, 2019*).

Of note, some studies highlight that Asian, Black, Hispanic, and multi-racial undergraduate students score higher for depression than white students (*Eisenberg et al., 2013*). Many theories have attempted to account for these racial and ethnic disparities, such as the effects of structural racism on symptoms, diagnosis, treatment and access to care (*Kendi, 2019*) and the use of white populations as the baseline norm (*Legha and Miranda, 2020*; for review see *Conrad, 2022*). Additionally, studies based on data from the national survey on drug use and health have shown ongoing mental health disparities among people identifying as Black, Indigenous, and people of color (BIPOC), women, and lesbian, gay, bisexual, and queer (LGBQ+) people (*SAMHSA, 2021*). These populations are also particularly at risk in academia due to structural inequities, barriers, and microaggressions among other challenges (for example, academic cultural barriers and stressors for BIPOC, *Halsey et al., 2020*; barriers for women doctoral students, see *Carter et al., 2013*; LGBQ+ academics experience toxic environments and microaggressions, see *Beagan et al., 2021* and *Linley and Nguyen, 2015*) that can impact the mental health of these populations. To acknowledge the systemic barriers faced by historically excluded groups,

and to recognize that people of certain backgrounds were intentionally excluded from parts of American society for various political, economic, and social reasons (e.g., *Berhe et al., 2021*; *Rollnick, 2015*) the terms 'historically excluded by race/ethnicity, gender, or sexual orientation' will be used to refer to these groups respectively. While the list of historically excluded groups is not comprehensive, we aim to decenter Whiteness by referring to historically excluded (HE) versus non-historically excluded (NHE) groups.

The COVID-19 pandemic has resulted in globally increased symptoms of anxiety, depression, post-traumatic stress disorder and psychological stress, particularly among healthcare workers, those with pre-existing mental health conditions, women, college students, and individuals under 45 (*Xiong et al., 2020*). Furthermore, it may also have impacted mental health during academic training (*Byrom, 2020*). Compounding systemic inequities and racial injustice, COVID-19 disproportionally impacted communities of color in the United States (*Li, 2020*; *Webb Hooper et al., 2020*; *CDC, 2020*). In particular, the highest risk of age-adjusted mortality during COVID was identified for Hawaiian and other Pacific Islander, American Indian or Alaska Native, and Latinx or Hispanic people (*Feldman and Bassett, 2021*).

While students themselves may be classified as associated with some lower risk groups for direct effects of COVID due to their levels of education, families of students identifying as BIPOC may be disproportionately affected, especially when intersecting with risk factors associated with social class (e.g., *Pathak et al., 2021*; *Feldman and Bassett, 2021*). Hence, students historically excluded because of their race and ethnicity (also termed persons excluded due to ethnicity or race, see *Asai, 2020*) may experience disproportionate impact and heightened concerns about individual, community and family health (*Pathak et al., 2021*; *Feldman and Bassett, 2021*; *Limas, 2021*; *Blake et al., 2021*). Indeed, a recent longitudinal study on undergraduates at the University of North Carolina at Chapel Hill (UNC-Chapel Hill; same location as the present study) found that the prevalence of anxiety and depression increased among first-year undergraduate students of color, sexual minority students, and women-identifying students during the pandemic (*Fruehwirth et al., 2021*).

Moreover, the stress caused by the COVID-19 pandemic coincided with heightened responses and protests against the persistent racial injustice found in the United States – notably killings of Black Americans by police and vigilantes. These

also took place alongside heightened anti-Asian violence, which may have also affected the mental health of Asian American students within academia.

At UNC-Chapel Hill, a large public university in the United States, the pandemic impacted courses and degrees in different ways. Most biomedical doctoral programs require didactic classes in the first two years only; afterwards students require full-time laboratory access to continue their research for another three to four years, precluding a transition to fully online instruction. Due to health and safety measures, biomedical doctoral student training was severely curtailed at our institution in the spring of 2020, and students returned to labs in June 2020 with strict occupancy limits. In contrast, medical student training is characterized by two years of didactic coursework and frequent test-taking, followed by another two years of sequential clinical experiences in hospitals or other clinical settings. In 2020, medical students could continue their academic progress with online learning options fostered by a switch to remote instruction in place of didactic in-person courses and clinical rotations. Hence, these students were temporarily removed from clinical rotations due to shortages of personal protective equipment; conversely though, graduation dates were not delayed.

Given this background, we sought to examine effects of the concurrent COVID-19 pandemic and heightened community reactions to racial injustice between 2019 and 2020 by exploring the mental health of medical students and biomedical doctoral students at UNC-Chapel Hill. We investigate comparisons between historically excluded (HE) and non-historically excluded (NHE) groups in science, focused on race/ethnicity, gender, and sexual orientation in medical and graduate biomedical students during the intersection of the COVID-19 pandemic and a time of reckoning of persistent racial injustice. Recognizing that an intersectional approach encompasses a multitude of identities (*Cho et al., 2013*), the facets of identity explored in this study are not intended to be comprehensive, but rather to represent major subpopulations represented in our sample and commonly found in US biomedical graduate and medical education.

## Methods

The UNC-Chapel Hill School of Medicine annually enrolls approximately 800 medical and 600 biomedical PhD students; part of the student body is representative of diverse identities including trainees from a variety of racial and ethnic groups, genders and LGBQ +identities. For example, a recent report (*UNC School of Medicine Office of Diversity Equity and Inclusion, 2021*) indicated that the medical student population at UNC-Chapel Hill consisted of 54% women, 58% white, 14% Asian, and 24% underrepresented students (4% not reported); the biomedical graduate student population consisted of 57% white, 15% Asian, and 26% underrepresented students (2% not reported). The present study was reviewed and approved by the Institutional Review Board (#18–0112).

The annual School of Medicine survey used for this study includes self-reported mental health status (examined in this work), as well as assessment of current and desired student-support programs. It is conducted to inform current and future mental health and wellness programming and global satisfaction with services provided (https://doi.org/10.17605/OSF.IO/H9UCX). The same School of Medicine survey was administered to medical and biomedical doctoral students approximately one year apart in 2019 and 2020 (n=431, Fall 2020; n=526, Fall 2019). Each survey was open for four weeks between September and October and distributed using the same mechanisms (e.g., same internal listservs) to be as comparable as possible.

In the primary analysis, mental health data was analyzed by type of training program (MD vs. PhD; that is, medical students vs. biomedical doctoral students), year (2019 vs 2020), and historically excluded vs. non-historically excluded (HE vs NHE) populations based on race/ethnicity (HE-RE vs. NHE-RE) and gender (HE-G vs. NHE-G, that is students identifying as women vs. men). A post hoc analysis included the primary variables as well as historically excluded on the basis of sexual orientation (HE-SO vs. NHE-SO, that is LGBQ+ vs. non-LGBQ+ populations).

### Measures

Demographic data including race/ethnicity were collected. In some cases, partial survey data was recorded (n=957 total responses; n=931 completed the survey; of those, n=740 submitted some or all demographic data). Partial surveys were used, however, only fully completed measures were included in the analysis (a blocked survey design enabled a data cleaning check that ensured participants completed each section/measure they were working on before closing the survey).

We assessed four measures of mental health (depression, anxiety, hazardous alcohol use, and problems related to substance use) using widely utilized and validated questionnaires. For each of these four measures, the values of responses were summed and sorted into categories of increasing severity ranging from 0 (indicating a lack of presentation of symptoms/no problematic substance use) to 3 or 4 (indicating increasing population-normed levels of severity for each variable), and then recoded into no symptoms (0) versus any symptoms (1). Depression scores/categories were calculated using the Patient Health Questionnaire (PHQ-9; *Kroenke et al., 2001*); anxiety scores/categories were calculated using the Generalized Anxiety Disorder Assessment (GAD-7; *Spitzer et al., 2006*); hazardous alcohol use was assessed using the Alcohol Use Disorder Identification Test (AUDIT; *World Health Organization, 2001*); problems with drug use was assessed by the Drug Abuse Screening Test (DAST-10; *Skinner, 1982*; *Yudko et al., 2007*; see *Table 1* for details of the categories and how they were created). To assess meaningful categories of symptomatic versus asymptomatic responses, each mental health outcome measure (for depression, anxiety, alcohol use and drug use) was recoded into bivariate (0/1) variables with a value of one (1) indicating symptoms or problems with each category (see *Table 1*).

In addition, suicidal ideation was assessed using the following three "Yes" or "No" questions: "Have you ever thought about ending your life?"; "Have you ever thought about ending your life while enrolled?"; "Have you ever thought about ending your life in the last 12 months?".

### Participants

All of the approximately 800 medical students and 600 biomedical doctoral students on campus were invited to participate in the survey each year via student listservs. Respondents included both medical and biomedical doctoral students (Fall 2020 n=431, Fall 2019 n=526, n=957 total respondents). This sample included 622 medical students, 309 biomedical doctoral students, with 26 surveys missing data, for a total of 931 completed surveys. Amongst those 931 respondents, 91 medical students were classified as HE-RE and 531 as NHE-RE; and 57 biomedical doctoral students were classified as HE-RE, with 252 being classified as NHE-RE.

Ages ranged from 18 to 40+, with the majority of students (59%) being ages 21–25 years (59% of medical students, 60% of biomedical doctoral students), followed by 35% being ages 26–30 (34% of medical students, 35% of biomedical doctoral students), and the remainder 5% being ages 31–35 (6% of medical students, 5% of biomedical doctoral students), with <1% each respectively (for both medical students and biomedical doctoral students) for ages 18–20 years and 35–40+years.

Respondents classified as belonging to NHE-RE groups identified as follows: 66% white (67% of medical students, 66% of biomedical doctoral students), 14% Asian (16% of medical students, 9% of biomedical doctoral students), or 2% other (2% of medical students, 3% of biomedical doctoral students). In accordance with National Institutes of Health definitions of 'underrepresented' in the biomedical, clinical, behavioral, and social sciences (*National Institutes of Health, 2020*), respondents were classified as belonging to HE-RE groups if they identified as follows: 7% African American (7% of medical students and biomedical doctoral students), 7% Latinx (4% of medical students, 11% of biomedical doctoral students), 2% Middle Eastern (3% of medical students, 1% of biomedical doctoral students), 1% American Indian/Alaskan Native (1% of medical students and biomedical doctoral students), and <1% Pacific Islander (<1% of medical students and biomedical doctoral students). Historically, Asian Americans have also faced exclusion from American culture, as evidenced by the Chinese Exclusion Act and the internment of Japanese Americans during the Second World War. However, Asian Americans have been well-represented in the sciences and thus are included as NHE-RE for analysis. Additionally, Middle Eastern is not an identity included in the United States Census or NIH definitions as underrepresented; however, this group is included as HE-RE due to the marginalization this population as experienced in the United States, especially in recent years and specifically over the decades following 9/11/2001 (e.g., *Crawford et al., 2021*; *Daraiseh, 2012*; *Clay, 2011*).

The majority of respondents identified as women (67%; 63% of medical students, 75% of biomedical doctoral students), followed by men (32%; 36% of medical students, 24% of biomedical doctoral students), and <1% other (genderqueer, gender nonconforming, gender non-binary, and transgender; equally represented across medical and biomedical doctoral students). Institutionally, the overall medical and biomedical doctoral student populations include more women than men, so these distributions

**Table 1.** Coding of the four measures of mental health used in the analyses.

Each measure of mental health (depression, anxiety, hazardous alcohol use, and problems related to substance use) was examined and coded according to its respective validated scale (PHQ-9, GAD-7, AUDIT, DAST-10). The scores were recoded as interim measures based on symptom severity, from 0 to 4. These were then transferred into a clinically meaningful bivariate category (no symptoms, 0; any symptoms, 1) to facilitate the planned analysis (bivariate logistic regression).

| Measures | No Symptoms (Coded as 0) | | Any Symptoms (Coded as 1) | | |
|---|---|---|---|---|---|
| | Interim Recoded as 0 | Interim Recoded as 1 | Interim Recoded as 2 | Interim Recoded as 3 | Interim Recoded as 4 |
| PHQ-9 | 1–4: Minimal or No Depression | 5–9: Mild Depression | 10–14: Moderate Depression | 15–19: Moderately Severe Depression | 20–27: Severe Depression |
| GAD-7 | 0–4: Minimal Anxiety | 5–9: Mild Anxiety | 10–14: Moderate Anxiety | 15–21: Severe Anxiety | |
| AUDIT | 0–7: Low Risk | 8–15: Hazardous | 16–19: Harmful | 20+: Possible Dependence | |
| DAST-10 | 0: No problems | 1–2: Low Problems | 3–5: Moderate Problems | 6–8: Substantial Problems | 9–10: Severe Problems |

are not unexpected. Women were classified as historically excluded and men as non-historically excluded. Though transgender individuals also experience mental health and substance use disparities (*National Center for Transgender Equality, 2016*), we were unable to include analyses in this study due the small sample size; future studies focused on the mental health and wellbeing of gender diverse people, including transgender and gender non-conforming individuals are needed.

Respondents were classified as non-historically excluded on the basis of sexual orientation (NHE-SO) if they identified as straight/heterosexual (83%; 85% of medical students, 79% of biomedical doctoral students); or historically excluded on the basis of sexual orientation (HE-SO) if they identified as bisexual (7%; 5% of medical students, 10% biomedical doctoral students), gay/lesbian (5%; 5% of medical students, 4% of biomedical doctoral students), queer (2%; 3% of medical students, 2% of biomedical doctoral students), pansexual (2%; 1% of medical students, 3% of biomedical doctoral students), asexual (1%;<1% of medical students and biomedical doctoral students), or other (1%; <1% of medical students, 2% of biomedical doctoral students).

### Analysis plan

A logistic regression was conducted, including significant interaction terms, to examine the effects of each variable on mental health outcomes using a parsimonious model. Full models with main effects and interactions were run for each variable, with stepwise addition of the largest interaction terms to the main effects, until the next largest interaction term added was no longer significant. At that point, the prior model with significant main effects and any significant interactions was retained as the final model for that variable (see *Source data 1*).

As previously described, each mental health outcome variable was split into clinically meaningful bivariate categories for depression, anxiety, problems with drug use and hazardous alcohol use such that the baseline category (none or fewest symptoms, as defined by each scale) was coded as zero (0), and any symptoms, as defined by more or worse symptoms than the baseline category, were coded as one (1). A bivariate logistic regression model was used to assess the impact of our primary factors of interest across the medical and biomedical doctoral student populations to maintain a large sample size and sufficient power to compare historically excluded (HE) and non-historically excluded (NHE) groups pre- and during COVID-19; groups were analyzed via Race/Ethnicity (RE) and Gender (G) for the primary analysis.

Year was coded into a practically meaningful bivariate category, with pre-COVID-19 as zero (2019; 0) and during the COVID-19 pandemic and racial unrest as one (2020; 1). Finally, HE and NHE statuses (based on race/ethnicity, gender, and sexual orientation) were coded into practically meaningful bivariate categories, with

endorsement of any historically excluded category being coded as one (1), whereas all others were coded as NHE using the value of zero (0).

The primary analysis included type of program (MD vs. PhD) x Year x Race/Ethnicity x Gender. A post hoc model included all these variables, with the addition of sexual orientation (all three historically excluded social identity groups; HE-RE, HE-G, & HE-SO).

### *Limitations*

Response bias is always a consideration in cross-sectional self-report research. Relatedly, there were no matched controls, hence it is possible that sampling distributions may have differed by chance. Furthermore, it is possible that respondents differed based on how important mental health is to them, potentially skewing the sample; hence we cannot definitively evaluate the respondent sample as representative of the full population. Due to anonymous data collection and optional questions to protect participant identities, we cannot assess granular response rates by demographic characteristics. Future studies could be completed with a controlled sample matched with participant identifiers to know who in the sample is and is not responding, as this could impact findings. Nonetheless, we achieved a response rate commensurate with voluntary survey data, suggesting a typical level of participation. We also had a high percentage of women respondents, who experience higher rates of depression and anxiety compared to men respondents; however, greater response from women is not atypical of our graduate medical and biomedical doctoral student populations, which include more enrolled women than men.

While anonymized data collection was employed as a strategy to increase sample size (e.g., more participants would feel safe responding if they could be anonymous), another limitation introduced as a byproduct of this design was the inability to examine the extent of repeated participants from year to year, which would be ideal to assess and control for in future studies. Nonetheless, using a between-groups design assuming normal variation in participants, there is no reason to believe that the two samples collected should be atypical from their respective populations over the consecutive years sampled. Methodological congruence of data collection was employed to reduce systematic bias in responses from year to year. Hence, results should be interpreted with caution as we cannot assess the percentage of repeated participants from year to year, though we believe the samples to be representative of the populations at each timepoint.

Conversely, we recognize that a large number of participants responding in both samples could be problematic for assumptions of independence for the use of parametric statistics and logistic regression analysis. Because we collected data anonymously to protect respondents' privacy, we cannot assess to what extent respondents may have participated in both years, hence results should be interpreted with caution. It would be preferable to have identifiable data and be able to use a repeated-measures design to reduce error variance. Yet, to the extent that significant findings were achieved even with the increased error variance inherent in a between-subjects design, these results likely had large enough effect sizes to be identified even given the loss of power from using a between-subjects design rather than the preferred, more sensitive within-subjects design.

Additional limitations included our lack of ability to control for other possible variables of interest such as pandemic-specific factors, years in training, and departmental affiliation or specialty area. Pandemic-specific questions were not asked because we used the same annual survey questions in 2019 and 2020 to maintain comparable responses. Department and specialty information were not asked in order to protect anonymity, to increase comfort with responding, and to maximize response rates. Yet, populations were purposefully defined by reasonably homogenous training experiences into the two major clusters of interest: medical training and biomedical doctoral training. A large portion of the sample did not complete optional demographic questions (such as number of years in training), limiting our ability to include this in the analysis. Future work should consider controlling for as many of these variables as possible.

### Results

The logistic regression model identified significant differences on student mental health outcomes for depression and anxiety by year (interactions), as well as by program (biomedical doctoral students scored worse than medical students), and for those historically on the basis of race/ethnicity (HE-RE) and by gender (HE-G) (*Figure 1A*).

In the combined population (when medical and biomedical doctoral students are considered together) there was a decrease in depression

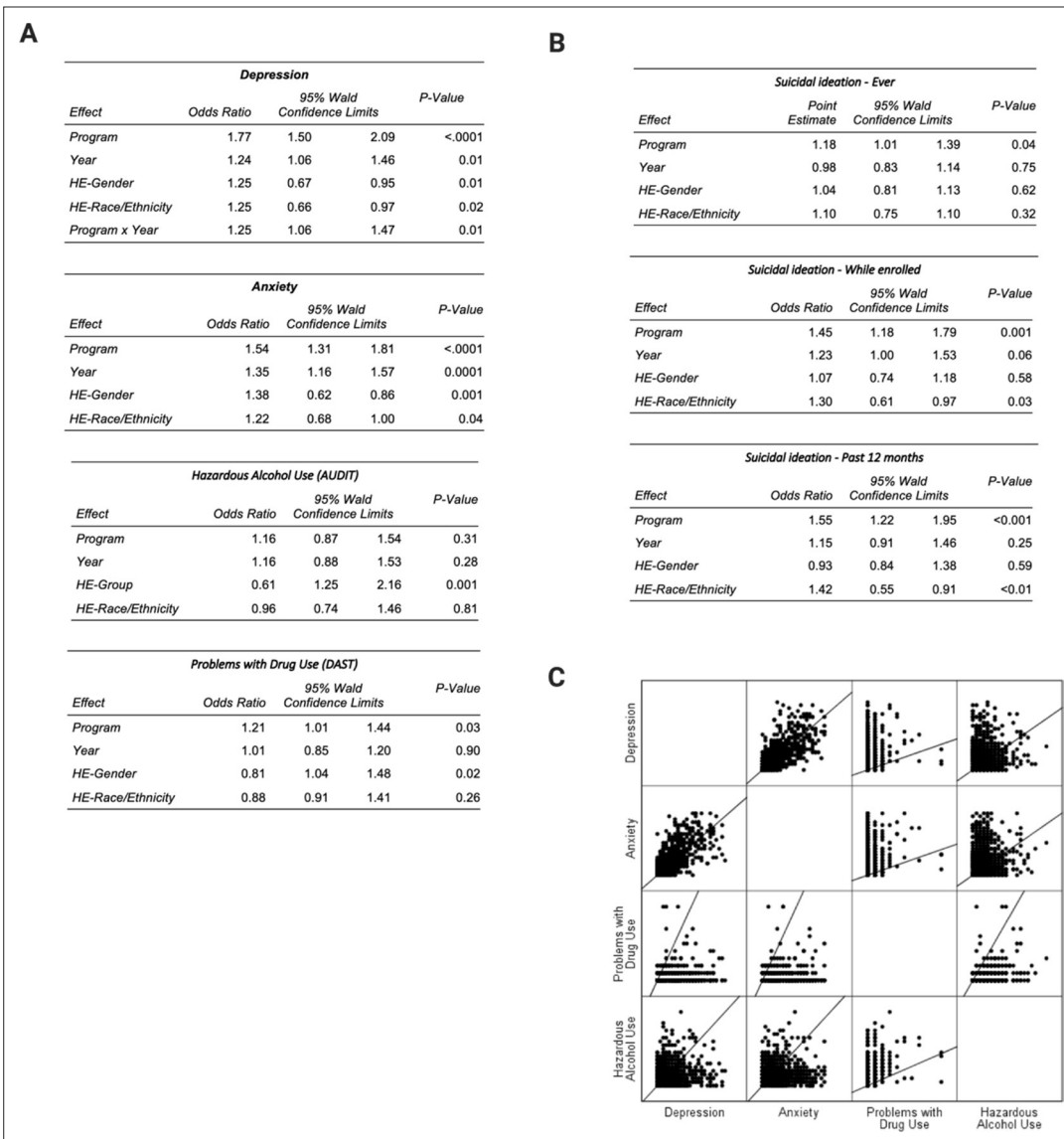

**A**

**Depression**

| Effect | Odds Ratio | 95% Wald Confidence Limits | | P-Value |
|---|---|---|---|---|
| Program | 1.77 | 1.50 | 2.09 | <.0001 |
| Year | 1.24 | 1.06 | 1.46 | 0.01 |
| HE-Gender | 1.25 | 0.67 | 0.95 | 0.01 |
| HE-Race/Ethnicity | 1.25 | 0.66 | 0.97 | 0.02 |
| Program x Year | 1.25 | 1.06 | 1.47 | 0.01 |

**Anxiety**

| Effect | Odds Ratio | 95% Wald Confidence Limits | | P-Value |
|---|---|---|---|---|
| Program | 1.54 | 1.31 | 1.81 | <.0001 |
| Year | 1.35 | 1.16 | 1.57 | 0.0001 |
| HE-Gender | 1.38 | 0.62 | 0.86 | 0.001 |
| HE-Race/Ethnicity | 1.22 | 0.68 | 1.00 | 0.04 |

**Hazardous Alcohol Use (AUDIT)**

| Effect | Odds Ratio | 95% Wald Confidence Limits | | P-Value |
|---|---|---|---|---|
| Program | 1.16 | 0.87 | 1.54 | 0.31 |
| Year | 1.16 | 0.88 | 1.53 | 0.28 |
| HE-Group | 0.61 | 1.25 | 2.16 | 0.001 |
| HE-Race/Ethnicity | 0.96 | 0.74 | 1.46 | 0.81 |

**Problems with Drug Use (DAST)**

| Effect | Odds Ratio | 95% Wald Confidence Limits | | P-Value |
|---|---|---|---|---|
| Program | 1.21 | 1.01 | 1.44 | 0.03 |
| Year | 1.01 | 0.85 | 1.20 | 0.90 |
| HE-Gender | 0.81 | 1.04 | 1.48 | 0.02 |
| HE-Race/Ethnicity | 0.88 | 0.91 | 1.41 | 0.26 |

**B**

**Suicidal ideation - Ever**

| Effect | Point Estimate | 95% Wald Confidence Limits | | P-Value |
|---|---|---|---|---|
| Program | 1.18 | 1.01 | 1.39 | 0.04 |
| Year | 0.98 | 0.83 | 1.14 | 0.75 |
| HE-Gender | 1.04 | 0.81 | 1.13 | 0.62 |
| HE-Race/Ethnicity | 1.10 | 0.75 | 1.10 | 0.32 |

**Suicidal ideation - While enrolled**

| Effect | Odds Ratio | 95% Wald Confidence Limits | | P-Value |
|---|---|---|---|---|
| Program | 1.45 | 1.18 | 1.79 | 0.001 |
| Year | 1.23 | 1.00 | 1.53 | 0.06 |
| HE-Gender | 1.07 | 0.74 | 1.18 | 0.58 |
| HE-Race/Ethnicity | 1.30 | 0.61 | 0.97 | 0.03 |

**Suicidal ideation - Past 12 months**

| Effect | Odds Ratio | 95% Wald Confidence Limits | | P-Value |
|---|---|---|---|---|
| Program | 1.55 | 1.22 | 1.95 | <0.001 |
| Year | 1.15 | 0.91 | 1.46 | 0.25 |
| HE-Gender | 0.93 | 0.84 | 1.38 | 0.59 |
| HE-Race/Ethnicity | 1.42 | 0.55 | 0.91 | <0.01 |

**C**

**Figure 1.** Logistic regression summary tables and correlation scatterplot matrix. Logistic regression tables show main effects and interactions for each of the primary factors (Program, Year, HE-Gender, HE-Race/Ethnicity) across four measures of mental health (Depression, Anxiety, Hazardous Alcohol Use, Problems with Drug Use) and three measures of suicidal ideation (Ever, While Enrolled, In The Last 12 Months); the correlation matrix scatter plots illustrate the relationships between the four mental health measures. All percentages were calculated out of total valid responses; see *Figure 2* for percentages. Mental health outcomes were coded into a bivariate 0/1 indicating the absence or presence of the respective symptoms for depression and anxiety (see Measures). For suicidal ideation, response options included 'Yes' (indicating suicidal ideations) or 'No' (indicating no suicidal ideations) for each of the three categories (see Methods for exact question wording). (**A & B**) Logistic Regression. A significant odds ratio greater than 1 indicates that the target category is more likely than the comparison group to indicate a higher score on that variable, whereas a significant odds ratio of less than 1 indicates that the target category is less likely than the comparison group to indicate a higher score on that variable. Nonsignificant effects suggest odds ratios do not show a difference greater than expected by chance (~1). If the 95% confidence interval includes 1, then the odds ratio does not differ more than expected by chance (e.g., OR = 0.98, CI95%=0.83–1.14 means there is not a significant effect of Year for Suicidal Ideation Ever); if the confidence interval does not include 1, then it differs more than expected by chance. (**C**) Scatterplots of the relationship between variables of interest (Depression, Anxiety, Problems with Drug Use and Hazardous Alcohol Use) displayed include graphical representations in a matrix format.

and anxiety, likely driven by the medical student improvements; see Discussion. In general, HE-RE students exhibited significantly higher rates of depression and anxiety at about 1.5 times the rate of their NHE-RE peers across the combined medical and biomedical doctoral student sample. No differences were evidenced on either drug or alcohol use by year, type of program or historically excluded/non-excluded status.

Overall, across medical and biomedical doctoral students, HE-RE students were approximately twice as likely to say they had thought about ending their life in the last 12 months than their NHE-RE peers, and about 1.8 times more likely while enrolled (*Figure 1B*); such high rates are extremely concerning. Due to different patterns emerging for training type and racial/ethnic identity by year, examining effect solely by year obscured these differences, hence interactions were further explored.

### Depression and anxiety

Surprisingly, for medical students, there was a significant decrease in depression and anxiety between 2019 and 2020 (*P*<.001), whereas doctoral students observed no change between years. However, HE-RE students differed from their NHE-RE peers, being significantly more likely to be depressed (*P*<.02) and anxious (*P*<.04). Women (HE-G) exhibited higher depression and anxiety scores than men (NHE-G) consistent with known mental health trends; hence these effects are controlled for in the analyses.

### Problems with drug use and hazardous alcohol use

No significant differences of note emerged between populations or within populations (e.g., NHE/HE by Program, Race/Ethnicity, Gender) on the primary measures. Reported problems with substance use and hazardous alcohol use were comparatively low in contrast to depression and anxiety, which were more pervasive. Men (NHE-G) exhibited more problems with drug use and hazardous alcohol use than women (HE-G) consistent with known substance use trends; hence these effects are controlled for in the analysis.

### Suicidal ideation

Medical students showed trends toward improvement, whereas biomedical doctoral students exhibited no change between 2019 and 2020. As compared with NH-RE peers, there were significant increase in HE-RE student suicidal ideation 'while enrolled' (*P*=.03), and 'in the last 12 months' (*P*<.01). There were no significant effects of gender on suicidal ideation (*p*s = .58 -.62, not significant).

### Medical school versus biomedical doctoral training

Prior to 2020, both medical and biomedical doctoral students suffered from depression (46% of medical students, 65% of biomedical doctoral students) and anxiety (47% of medical students, 67% of biomedical doctoral students) at high rates (*Figure 2A and B*), as defined by no symptoms compared with any symptomatic categories (see Methods for categorical definitions). Doctoral student mental health in 2020 remained very poor (depression: 26% of medical students, 64% of biomedical doctoral students; anxiety: 32% of medical students, 61% of biomedical doctoral students), whereas, surprisingly, the mental health of medical students improved. Suicidal ideation 'in the last 12 months' among biomedical doctoral students (compared with medical students) was markedly higher, both before (11% of medical students, 16% of biomedical doctoral students) and during 2020 (6% of medical students, 19% of biomedical doctoral students).

We assessed differences between medical and biomedical doctoral students for depression (PHQ-9), anxiety (GAD), problematic drug use (DAST), and hazardous alcohol use (AUDIT) using nominal outcome variables (*Figure 2A and B*). We identified significantly higher rates of depression and anxiety for biomedical doctoral students compared with medical students, as well as higher rates of suicidal ideation 'while enrolled' and 'in the last 12 months'. No significant differences emerged for problems with drug use or hazardous alcohol use by training type, year, or historical exclusion by race/ethnicity; furthermore, problems with drug use rates were comparatively low in contrast to depression and anxiety. Known gender effects were consistent with expectations for both problematic substance use categories (greater use found in prior studies is also reflected in our sample for NHE-G vs. HE-G).

### Historically excluded versus non-historically excluded students based on race/ethnicity (HE-RE vs. NHE-RE)

Mental health results for groups historically and non-historically excluded on the basis of race/ethnicity (HE-RE vs. NHE-RE) across a

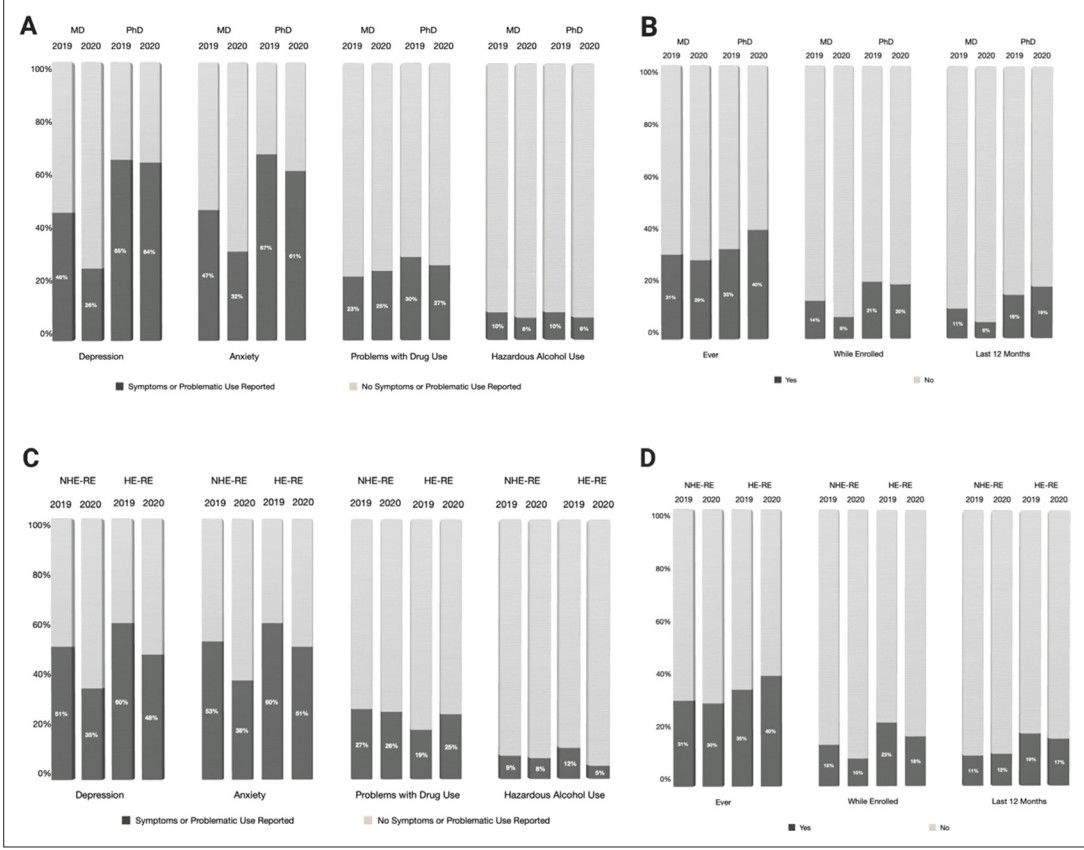

**Figure 2.** Levels of depression, anxiety, problems with drug use, hazardous alcohol use and suicidal ideation based on type of training and historically excluded status linked to race/ethnicity. (**A**) Percentages of medical (MD) and biomedical doctoral (PhD) students reporting the presence (dark grey) or absence (light grey) of symptoms/problematic use pertaining to depression, anxiety, problems with drug use and hazardous alcohol use. (**B**) Percentages of medical (MD) and biomedical doctoral (PhD) students in 2019 and 2020 reporting having had (dark grey) or not having had (light grey) suicidal thoughts ever, in the last 12 months or while enrolled. (**C**) Percentages of historically excluded and non-historically excluded students based on race and ethnicity (HE-RE vs HE-NRE) reporting the presence (dark grey) or absence (light grey) of symptoms and problematic use pertaining to depression, anxiety, problems with drug use and hazardous alcohol use. HE-RE students showed higher rates of depression in both 2019 (60% of HE-RE, 51% of NHE-RE) and 2020 (48% of HE-RE, 35% of NHE-RE). This pattern was also seen for anxiety (in 2019: 60% of HE-RE, 53% of NHE-RE; in 2020: 51% of HE-RE, 38% of NHE-RE). (**D**) Percentages of historically excluded and non-excluded students based on race and ethnicity (HE-RE vs HE-NRE) reporting, in 2019 and 2020, having had (dark grey) or not having had (light grey) suicidal thoughts ever, in the last 12 months or while enrolled. Historically excluded students exhibit higher rates of suicidal ideation, in particular 'while enrolled' and 'during the last 12 months' (2019, positive responses for 'while enrolled': 23% of HE-RE, 15% of NHE-RE; positive responses for 'during the last 12 months': 19% HE-RE, 11% NHE-RE; 2020, positive responses for 'while enrolled': 18% of HE-RE, 10% of NHE-RE; positive responses for 'during the last 12 months' 17% HE-RE, 12% NHE-RE). All percentages were calculated out of total valid responses. HE-RE students were coded as such if they indicated that they belonged to historically excluded racial or ethnic categories (e.g., African American/Black, Hispanic/Latinx; see Methods for details); NHE-RE included students who did not indicate a marginalized racial/ethnic identity.

combined pool of biomedical doctoral students and medical students were also compared (*Figure 2C and D*). In both 2019 and 2020, HE-RE students experienced higher rates of depression and anxiety compared to their NHE-RE peers (*Figure 2C and D*). Suicidal ideation was also worse for HE-RE students as compared with their NHE-RE peers, specifically 'while enrolled' and 'in the last 12 months'. In summary, HE-RE outcomes in general were worse for depression and anxiety (*Figure 2C*), as well asthese students experiencing more suicidal ideation 'while enrolled' and 'over the last 12 months' (*Figure 2D*).

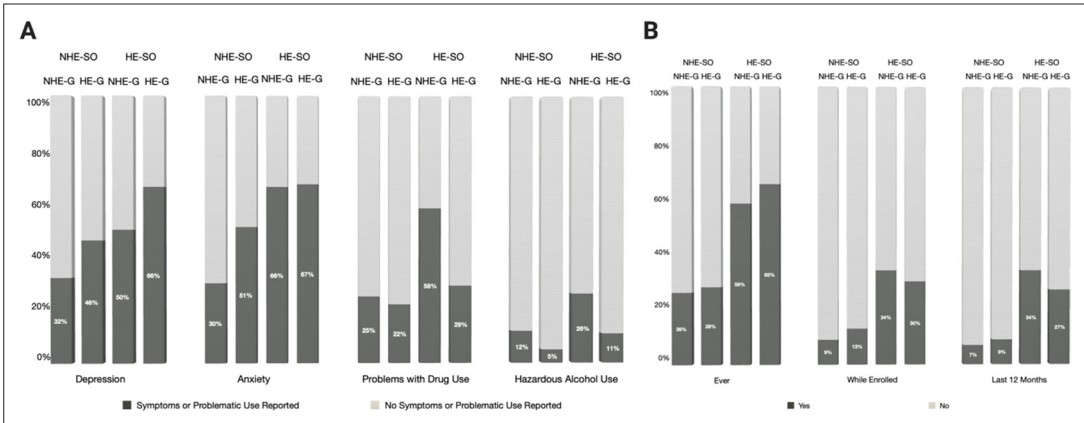

**Figure 3.** Levels of depression, anxiety, problems with drug use, hazardous alcohol use, and suicidal ideation for historically excluded/non-excluded populations based on gender and sexual orientation. Graphical representation displaying the percentages for each of the two primary factors in this figure (Gender and Sexual Orientation) across the four measures of mental health and three measures of suicidal ideation for populations historically excluded on the basis of sexual orientation or gender (HE-SO and HE-G). All percentages were calculated out of total valid responses. Historically excluded versus non-historically excluded students based on sexual orientation (HE-SO vs. NHE-SO) were compared. Members of the HE-SO group were classified as such if they indicated an excluded identity (e.g., Lesbian, Bisexual, Queer – LGBQ+ sexual orientation; see Methods for details); NHE-SO included any students who did not indicate a historically excluded identity. Historically excluded on the basis of gender included women; non-historically excluded students by gender included men (HE-G vs. NHE-G). (**A**) Percentages of LGBQ+ (HE-SO) and non-LGBQ+ (NHE-SO) men and women (NHE-G vs. HE-G) reporting the presence (dark grey) or absence (light grey) of symptoms and problematic use pertaining to depression, anxiety, problems with drug use and hazardous alcohol use. (**B**) Percentages of LGBQ+ (HE-SO) and non-LGBQ+ (NHE-SO) men and women (NHE-G vs. HE-G) reporting having had (dark grey) or not having had (light grey) suicidal thoughts ever, in the last 12 months or while enrolled.

While individuals identifying as Asian are well-represented in the academic workforce, 2020 saw an increase in anti-Asian violence, which led us to conduct additional analyses to compare the mental health of Asian American and HE-RE students in 2019 and 2020. People who identified as of Asian descent generally scored better than other NHE-RE or HE-RE groups, indicating fewer mental health challenges on all four outcome variables and the three suicidal ideation variables. However, conditional odds ratios based on these models indicated some changes by year that may be associated with the negative psychosocial and environmental factors many Asian Americans experienced between 2019–2020. Accounting for the effect of race (3-level: HE-RE Asian, HE-RE Non-Asian, HE-NRE) while controlling for year, type of training and gender, Asian American participants showed greater mean depression scores ($P<.03$) and anxiety scores ($P<.04$) in 2020 compared with 2019; there was no change in either alcohol use ($P=.07$, though marginal) or drug use ($P=.28$). There were no significant changes in conditional probabilities for suicidal ideation of Asian American participants during that time.

## Associations

A robust positive association was evident between depression and anxiety ($r=.69$, $P<.001$), not surprisingly as these conditions are often comorbid. Both depression ($r=.14$) and anxiety ($r=.33$) were associated with problems with drug use ($ps <.001$). Neither depression ($r=-.01$, $P=.74$) nor anxiety ($r=-.01$, $P=.77$) were associated with hazardous alcohol use. Since this was only correlational in nature, a causal direction cannot be determined; nonetheless, these associations may indicate a connection between problems with drug use either as a precursor to or as an effect of experiencing mental health symptoms.

## Post hoc analysis: historically excluded versus non-historically excluded students based on sexual orientation (HE-SO vs. NHE-SO)

To further examine additional aspects of intersectionality, we completed a post hoc analysis including LGBQ+ identities (HE-SO) into a simplified model (controlling for main effects of the four primary variables and identifying any significant

two-way interactions). Due to a smaller sample size, this was included as a post hoc analysis.

The HE-SO variable showed significant main effects across main outcome variables, with HE-SO participants more than twice as likely to experience depression (OR = 2.19, $P<.0003$) and anxiety (OR = 2.66, $P<.0001$). In addition, there was a significant interaction with 'Year' for depression ($P=.03$), such that 2020 was significantly worse for HE-SO depression than 2019.

There were also main effects for alcohol use (OR = 2.31, $P<.01$) and drug use (OR = 2.37, $P<.0002$), such that HE-SO participants were more at risk for substance use; however, an interaction between gender and sexual orientation ($P=.03$) indicated that among HE-SO participants, NHE-G (LGBQ +men) were more likely to report drug use than HE-G participants (LGBQ+ women; conditional OR = .29, $P<.003$). HE-SO NHE-G participants (LGBQ+ men) compared to NHE-SO HE-G participants (LGBQ+ women) were nearly four times as likely to report drug use (conditional OR = 3.91, $P<.0002$), whereas HE-SO HE-G participants (LGBQ+ women) were not significantly more likely to report drug use ($P=.19$).

HE-SO participants experienced more suicidal ideation (ever, OR = 4.37, $P<.001$; while enrolled, OR = 3.66, $P<.0001$; in the last 12 months, OR = 4.77, $P<.0001$; see *Figure 3* for visualization of comparative percentages). In addition, there was a marginal interaction between 'HE-SO' and 'Program' ($P=.05$), such that medical participants were much less likely (OR = .25) than biomedical doctoral students to indicate suicidal ideations while enrolled. Hence, biomedical doctoral HE-SO participants were almost six times more likely to report suicidal ideations than NHE-SO biomedical doctoral participants while enrolled (conditional OR = 5.94, $P<.0001$) whereas HE-SO medical students where about twice as likely to indicate suicidal ideations while enrolled than NHE-SO medical students (conditional OR = 2.925, $P<.02$).

## Discussion

Overall, medical student mental health improved from 2019–2020 on measures of both depression and anxiety, whereas biomedical doctoral students showed no change, their scores remaining concernedly high. HE-RE students were worse off than their NHE-RE peers in measures of depression, anxiety, and suicidal ideation – particularly reporting more suicidal ideation than their counterparts both 'while enrolled' and in

the 'last 12 months'. All analyses controlled for gender, for which women (HE-G) generally experienced anxiety and depression more than men (NHE-G), whereas men generally reported higher substance use. Post hoc analyses also indicated that those identifying as HE-SO (LGBQ+ respondents) experienced more anxiety and depression than NHE-SO (non-LGBQ+) respondents, as well as greater suicidal ideation.

### Trends in medical students versus biomedical doctoral students

Despite unique challenges between 2019 and 2020, many metrics of medical student mental health improved. In mid-March of 2020, medical students were pulled from clinical settings and from typical coursework due to the pandemic. Usual academic and clinical responsibilities were replaced with a four-week online course named 'Medical Management of COVID-19', which focused for instance on wellness, self-care, medical management and personal protective equipment related to COVID-19 (*UNC Health and UNC School of Medicine, 2020*). It is possible that the pause in traditional medical student training and the new course focused on timely topics mitigated stress-induced mental health erosion. In contrast, biomedical doctoral students did not receive any similar interventions addressing emerging COVID knowledge or mental health coping strategies.

Furthermore, medical students at our institution were able to continue some form of their training remotely, to maintain some social contact and peer support networks through academic training, and to avoid graduation delays. Additionally, while medical students pay for their training, which could contribute to financial stress, these financial demands did not change between 2019 and 2020. This consistency of circumstances, combined with expectations of income post-graduation and a near-guarantee of eventual employment in medicine, could also have mitigated the downward trend in medical student mental health. Furthermore, we also speculate that being in a medical training program with obvious direct impact during a time when medical professionals were publicly celebrated (at least initially) had a positive effect on the mental health of medical students.

Finally, and in contrast to biomedical doctoral programs, medical school accreditation bodies mandate and enforce the presence of more structured mental health and wellness support systems, ensuring that support services for

medical students are more easily accessible and freely available (no cost, and/or available funding to cover any out-of-pocket mental health medical expenses). At our institution, both medical and biomedical doctoral students have limited access to facilitated and triaged on-campus resources; however, while the equivalent of 1.5 person working full time is dedicated to support the ~800 medical students (e.g., 533:1 student-counsellor ratio), the ~600 biomedical doctoral trainees only have access to the equivalent of 0.5 full-time employee dedicated to their support (e.g., 1200:1 student-counsellor ratio). The more robust support for medical students, combined with being removed from the acute stressors often encountered in rotations (due to being pulled from clinical duties during the pandemic) may have magnified the ameliorative effect of the cohort-based intervention that medical students received.

In contrast to medical students, biomedical doctoral students exhibited alarmingly high levels of depression and anxiety that dwarfed the rates in the medical student population, which were themselves higher than in the general population. This is similar to previous findings (e.g. *Evans et al., 2018*), even accounting for COVID trends (see *Gordon, 2021*). When considering recent suicidal ideation in particular, the mental health of biomedical doctoral student worsened between 2019 and 2020. This may be linked to changes imposed on biomedical doctoral student training: laboratory research had to be stopped and lab occupancy was then restricted in Spring 2020, which reduced supportive social interactions and peer camaraderie. The loss of progress on dissertation projects (given that graduation times are open-ended) was frustrating and may have exacerbated mental health issues. In addition, the concurrent economic recession may have depressed optimism about job opportunities for biomedical doctoral students, who pursue a much wider range of careers after graduation compared to medical students. The dismissal of science by politicians and the media, popular sentiments against public health policies based on scientific recommendations and public degradation of trust in science and scientists may have all contributed to a worsened experience for biomedical scientists (e.g., *Couée, 2020*; *Gross, 2020*; *Kreps and Kriner, 2020*).

For biomedical doctoral students whose research was not directly related to COVID-19, it may have been harder to find meaning and purpose in their formerly satisfying research on other diseases. Anecdotally, some biomedical doctoral students sought opportunities to volunteer their skills in molecular biology in diagnostic COVID-19 labs, indicating a desire to contribute to the immediate crisis. In contrast, medical students may have found it easier to connect their training with the real-world crisis of COVID-19. Moreover, when labs were shutting down, medical education shifted to COVID-19 prevention topics directly related to the pandemic.

Additionally, medical students will soon become medical doctors in a society where this profession took center stage. On the other hand, the success of the vaccine effort – which received praise but also backlash – may yet improve how basic research scientists are considered in society in the coming years (*Bogel-Burroughs, 2020*; *Kauer, 2020*; *Kolata, 2021*). Future education and mental health research should delve into the aspects of medical education versus biomedical doctoral training that drive different mental health outcomes during a crisis and in normal times.

## Mental health trends for student from historically excluded racial or ethnic groups

Irrespective of medical or doctoral program, mental health metrics for depression, anxiety and suicidal ideation were worse for students from historically excluded groups based on race/ethnicity compared to students from non-historically excluded racial groups, and they worsened between 2019 and 2020. Overall, the incidence of suicidal thoughts for HE-RE students increased proportionately both 'while enrolled' and 'in the last 12 months' compared with NHE-RE students, and particularly for biomedical doctoral HE-RE students 'in the last 12 months'. We hypothesize that the COVID-19 pandemic and heightened racial unrest contributed to these differences, although future research is needed to isolate the factors causing the observed effects. This result may be partially explained by the added stressors of systemic racism in academic, medical, and American cultures. Academic culture, in particular, is based on norms that systematically exclude minority groups and create additional mental health challenges for people of color (e.g., persons excluded due to ethnicity or race, see *Asai, 2020*).

## Associations

Our finding that depression was correlated with both anxiety and problematic drug use was consistent with previous work which demonstrates

common comorbidity of depression and anxiety, as well as of substance use disorders with mental illness (e.g., *National Institute on Drug Abuse, 2010*); hence, it is plausible that problematic drug use could represent a contributing factor and/or a coping strategy for depression but this should be interpreted with caution. Furthermore, the lack of association in our sample between hazardous alcohol use and mental illness was surprising given the common associations found between these variables in previous work (e.g., *Smith and Randall, 2012*; *McHugh and Weiss, 2019*).

### Mental health trends for students belonging to groups historically excluded because of sexual orientation

Our findings that LGBQ+ students experience worse mental health outcomes align with national studies showing that sexual minorities are approximately twice more likely than heterosexual people to experience mental health conditions such as anxiety, depression, suicide, and substance use (*Medley et al., 2016*; *SAMHSA, 2020*; *SAMHSA, 2021*). Similar to national trends, gender interacted with sexual orientation to indicate worse effects for LGBQ+ women for depression and anxiety, but worse substance use for LGBQ+ men. While our small sample suggests that these results should be interpreted with caution, the large effect size evident in our analysis was extremely concerning and indicates a crucial need for studies to further examine this population and how it could be better supported. In addition, our inability to evaluate these hypotheses for transgender and gender non-conforming students (due to an insufficient sample size to run a parallel analysis) was a limitation. This should certainly be evaluated in future studies, especially due to transgender people being at increased risk and incidence for mental health problems, as evidenced nationally (*National Center for Transgender Equality, 2016*).

### Conclusions and recommendations

Whereas medical schools have had mental health recommendations in place since 1992 (*AAMC Executive Council, 1992*; enforced by the Liaison Committee on Medical Education accreditation standards), biomedical doctoral programs have had no such parallel policies regarding mental health provisions nationwide. Our new data suggest a need for swift action to address the very urgent mental health needs of biomedical doctoral students both before and

during the COVID-19 pandemic, especially for students who have been traditionally excluded from the academy based on their gender, race/ethnicity, and sexual orientation. Furthermore, the data we present suggests that depression, anxiety, and suicidality for historically excluded medical and biomedical research students have been exacerbated during the ongoing COVID-19 pandemic and climate of heightened visibility of racial inequity. Future studies should explore additional disparities between historically and non-historically excluded students in addition to those highly concerning trends.

It is crucial, now more than ever, to provide mental health support both on campus and remotely to ensure that students have access to the mental health services they need (see *Krause and Harris, 2019*). On-campus mental health resources should reflect the diversity of the student body, including in terms of race/ethnicity and sexual orientation. Meanwhile, mental health resources, communities, and support groups are starting to emerge online (e.g., PhDBalance, TAE Consortium, RVoice).

While offering more resources is an important step, the increasing national demand for mental health services on university and college campuses may make it difficult to provide comprehensive mental health care to all students who need it (*Seppälä et al., 2020*). In addition, simply providing services and programs may not be enough due to inequitable access to these resources and systemic issues that negatively impact mental health outcomes. Preventative health measures should therefore be investigated at the graduate level, particularly regarding the learning environment. Prevention, wellness resources, LGBTQ+ Safe Zone, anti-racism and resiliency training as well as assessment of impact are crucial to reduce the acute need for mental health support, including substance use.

Given the higher indicators of distress, the factors driving the mental health crisis, particularly for BIPOC students, those from gender and sexual minorities, and biomedical doctoral students, should be identified. Some potential causes to explore include toxic work environments, systemic racism, sexism, homophobia and unhealthy cultural and academic norms. In addition, the following factors contribute to historical exclusion based on race and ethnicity: bullying and harassment, precarity of work contracts in higher education, science inequities based on barriers and limitations to diversity, inclusion, and accessibility, and a disproportionate effect of the competitive culture in academia on historically

excluded groups (*Limas et al., 2022*; *Limas, 2021*). Researchers must continue to examine how these and other factors may contribute to negative experiences for historically excluded groups particularly, and to extend this research to proposing, implementing and evaluating needed programmatic and policy changes empirically. This should also include evidence-based research focused on the mental health of faculty and staff to better cater to this population's needs, as well as to recognize and improve the support they provide to students (e.g., *Loissel, 2019*; *Loissel, 2020*).

While individual protective measures can alleviate some negative impacts of operating within a flawed system (e.g., the academic environment), systemic change must occur rather than relying on those most likely to be impacted to create change (*Halsey et al., 2020*). It is imperative that leaders in higher education use evidence-based quantitative and qualitative research to examine population trends, create visibility for lived experience, and ultimately identify and reduce causes of mental health problems rather than just treating symptoms when they emerge. Academic culture needs to be actively reformed by those in power to model, encourage, and sustain student, faculty and staff wellbeing.

### Study limitations and future directions

Limitations related to anonymous data collection include the inability to estimate exact response rates, percentage of repeated versus new respondents, or non-respondents who skipped entering demographic data. In addition, self-reported mental health status may be biased, and independent assessments by a clinician would provide greater accuracy. Importantly, given the observational nature of our data collection, many confounds could also not be accounted for which changed over the course of the year between data collection timepoints. Nonetheless we believe that our work provides a compelling starting point to further examine trends and emergent concerns in the mental health of medical and biomedical doctoral students. Even given the limitations of our dataset, the unexpectedly high rate of suicidal ideation is concerning: among respondents alone, nearly 40 trainees (16 medical students and 21 biomedical doctoral students) reported recent suicidal ideation in 2020.

Many segments of society in the United States have been intentionally excluded from social, economic, and cultural opportunities via law, policy, and cultural expectations in order for dominant groups to retain power and privilege. Some groups have explicitly and implicitly experienced more systemic barriers than others – for example, the legal discrimination against African Americans throughout American history, or the restriction of voting privileges to exclude women and non-white voters. Historically, this has been reinforced by social norms and cultural biases, even in situations when overt racial discrimination has been addressed. We recognize that race is a social construct, and that other social identity groups beyond race and ethnicity (*Rothman et al., 2020*) may also experience inequitable impacts of COVID-19. For instance, we noted that women and LGBTQ+ students also showed evidence of negative impacts on their mental health in the current study. We have controlled for differences by gender as well as examined effects for sexual orientation (see *Salerno et al., 2020*), but we recognize that these analyses are not comprehensive of all social groups experiencing inequities, and we acknowledge the impact of additional identities that we were not able to study such as, among others, international and undocumented status (*Hunt, 2020*; *Chen et al., 2020*) and disability (*Goggin and Ellis, 2020*; *Gray et al., 2020*). In fact, the NIH has recently expanded its definition of underrepresented in science to include women, people with disabilities, first-generation college students and those from disadvantaged backgrounds (including people who have experienced or are experiencing homelessness, foster care participation, recipients of free and reduced lunch, Pell grants, SNAP or WIC, and those who grew up in a low-income or rural areas; *National Institutes of Health, 2020*; *National Institutes of Health, 2018*, rescinded and replaced in 2019; *National Institutes of Health, 2019*).

Yet this expanded definition remains flawed as some groups are still excluded from recognition and inclusion as societal norms and practices shift over time. The concept of 'historically excluded' should therefore continue to evolve and be re-examined or expanded over time as warranted by newly identified historical trends. For instance, people of Middle Eastern descent have faced heightened discrimination following 9/11/2001. Students identifying as Asian may also face distinct challenges that affect their mental health among rising anti-Asian violence (e.g., *Yam, 2021a*; *Yam, 2021b*). In fact, our analyses showed that the 2019–2020 year brought additional challenges that impacted mental health in a negative direction for both depression and anxiety in this

population, despite overall scores that indicate, on average, better mental health outcomes. This supports the need for examining Asian HE-RE populations separately in future work (and to contrast Asian American with white American experiences) in order to identify distinct experiences, protective factors and challenges. In sum, while not all groups nor all aspects of exclusion were represented in this paper, we aim to amplify the importance of examining multiple layers of identity and historical exclusion: future directions should include examination of the intersectionality of these and other identity groups, as well as systemic barriers that each may encounter differentially.

Furthermore, additional structural factors not accounted for herein (e.g., isolation, financial stressors, policy and law impacts) may have also exacerbated the mental health status of trainees and created systemic inequities for different groups. For instance, while data collection was completed before the 2020 election in the United States, political tensions were building throughout the summer and fall of 2020. Future research should examine the way that political cycles, economic trends and governmental policies impact trainee populations in the United States, and in particular how xenophobic, nationalist, ableist, sexist, and homophobic legislation affect the mental health of students from historically excluded communities.

## Acknowledgements
The authors thank the Odum Institute for Research in Social Science at UNC-Chapel Hill for its support, resources and consultations on statistical analyses. We thank Nathan Vanderford for valuable insights on earlier versions of this manuscript. Tables were generated from SAS Output, and figures were created using Numbers and BioRender.

**Allison Schad** is in the Office of Medical Education, University of North Carolina at Chapel Hill School of Medicine, Chapel Hill, United States
allison_schad@med.unc.edu
http://orcid.org/0000-0002-6672-5202

**Rebekah L Layton** is in the Office of Graduate Education, University of North Carolina at Chapel Hill School of Medicine, Chapel Hill, United States
rlayton@unc.edu
http://orcid.org/0000-0001-7113-1348

**Debra Ragland** is in the Office of Graduate Education, University of North Carolina at Chapel Hill School of Medicine, Chapel Hill, United States
http://orcid.org/0000-0001-9512-1988

**Jeanette Gowen Cook** is in the Office of Graduate Education and the Department of Biochemistry and Biophysics, University of North Carolina at Chapel Hill School of Medicine, Chapel Hill, United States
http://orcid.org/0000-0003-0849-7405

*Author contributions:* Allison Schad, Conceptualization, Resources, Data curation, Formal analysis, Funding acquisition, Investigation, Visualization, Methodology, Writing – original draft, Project administration, Writing – review and editing; Rebekah L Layton, Conceptualization, Resources, Data curation, Formal analysis, Supervision, Funding acquisition, Investigation, Visualization, Methodology, Writing – original draft, Project administration, Writing – review and editing; Debra Ragland, Validation, Writing – review and editing; Jeanette Gowen Cook, Resources, Funding acquisition, Writing – review and editing

*Competing interests:* The authors declare that no competing interests exist.

*Ethics:* Human subjects: The study was reviewed and approved as Exempt by the UNC Institutional Review Board (#18-0112). Consent was obtained on the first question of the survey before continuing forward to respond to any additional questions. In accordance with IRB approval, the presentation of data will not include demographic information that could potentially lead to the identification of a student. Furthermore, to protect student anonymity, responses with fewer than ten individuals will not be shared. Accordingly, data sharing is limited to those variables directly relevant to the analyses conducted (e.g., underlying data) and composite categories (e.g., historically excluded by category vs. non-historically excluded by category, such as race/ethnicity, gender, or sexual orientation) are shared when applicable rather than granular demographic information to protect the identity of respondents.

## Funding

| Funder | Grant reference number | Author |
| --- | --- | --- |
| National Institute of General Medical Sciences | 1R01GM140282-01 | Rebekah L Layton |

The funders had no role in study design, data collection and interpretation, or the decision to submit the work for publication.

## Decision letter and Author response
Decision letter https://doi.org/10.7554/eLife.69960.sa1
Author response https://doi.org/10.7554/eLife.69960.sa2

# Additional files

## Supplementary files
- Transparent reporting form
- Source data 1. Source code for SAS code and sample model.
- Reporting standard 1. STROBE checklist.

## Data availability
Underlying data is available on the Open Science Framework at https://doi.org/10.17605/OSF.IO/H9UCX. In accordance with IRB approval, the presentation of data does not include demographic information that could potentially lead to the identification of a student. Furthermore, to protect student anonymity, responses with fewer than ten individuals will not be shared. Accordingly, data sharing is limited to those variables directly relevant to the analyses conducted (e.g., underlying data) and composite categories (e.g., underrepresented vs. well-represented) are shared when applicable rather than granular demographic information to protect the identity of respondents also noted in in the Human Subjects section.

The following dataset was generated:

| Author(s) | Year | Dataset URL | Database and Identifier |
|---|---|---|---|
| Schad A, Layton RL, Ragland D, Cook JG | 2021 | https://doi.org/10.17605/OSF.IO/H9UCX | Open Science Framework, 10.17605/OSF.IO/H9UCX |

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
