## [Decision Letter]

**Decision letter after peer review:**

Thank you for submitting your article "Mental health disparities among biomedical trainees during COVID-19 and racial inequity" to *eLife* for consideration as a Feature Article.

Four peer reviewers have reviewed your article, and the evaluation has been overseen by two members of the *eLife* Features Team (Elsa Loissel and Peter Rodgers). The following individuals involved in reviewing your submission have agreed to reveal their identity: Marcus Lambert and Amanda Haage.

The reviewers and editors have discussed the reviews and we have drafted this decision letter to help you prepare a revised submission.

Please also note that if your manuscript is accepted, we would like to publish some of the relevant code with the article: please let me know if this will be a problem.

In addition, the *eLife* Features Editor may also contact you separately about some editorial issues that you will need to address.

Summary:

The article aims to explore the wellbeing of medical students and biomedical PhD students at an R1 institution using self-reported markers of mental health across race/ethnicity. This work was conducted over two years, one of which was marked by the COVID-19 pandemic and a wave of racial and civil unrest. This manuscript is important and timely for a broad biomedical audience, highlighting major concerns around the prevalence of mental health challenges, especially in biomedical PhD and underrepresented students.

However, several points need to be addressed to make the article suitable for publication. In particular, the reviewers were concerned with some aspects of the methodology and the discussion.

Essential revisions:

Comparing medical and PhD students:

1 – Please consider discussing in the introduction (and the discussion) why there may be a difference between medical and biomedical PhD students. In particular, please expand on:

a. The level of institutional support provided by the respective programs: For instance, what are the mental health resources provided to medical and PhD students? Are they the same and are they easily accessible e.g. paid for? facilitated/triaged? on-campus?

b. The awareness of the two populations of their mental health (one might argue that medical students are trained to recognize certain signs and symptoms).

c. The differences that are intrinsic to their training: for example, those familiar with the medical student training process might suggest more acute stressors but more structured support due to LCME requirements.

d. The response to the COVID pandemic from the two different programs: for instance, the curriculum and/or level of support changed for the medical students in 2020, which is highlighted in the discussion. However, this is a big limitation in interpreting the data (comparatively between MD and PhD students). This can be viewed as an intervention that the PhD students did not have.

2 – Following that thought: medical and graduate students have extremely different training paradigms and, in this institution, medical students received an intervention that could have mitigated the effects of the pandemic on their mental health. Therefore, please consider whether directly comparing medical to graduate students is relevant, and if so, how to do so: for example, the bar graphs side by side are fine, but the chi-sq tests may be an unfair comparison. Or at least, please discuss the caveat of the different training paradigms and levels of support provided throughout the 2020 year.

3 – The reason why PhD students have worse PHQ9 outcomes over time comparatively is because the medical students show a decrease from 2019 to 2020. Could you please report whether you have looked at the trends across the medical student years (1 through 4), and justify if not.

Discussing the sample, and how it was coded:

4 – Line 158: Please detail the criteria for sample invitations (e.g. are there any inclusion or exclusion criteria for the study, and if so, are they the same for the 2019 and 2020 surveys?).

5 – Line 158: Please clarify whether the respondents (n=431, Fall 2020; n=526, Fall 2019) are a representative sample that closely matches the characteristics of your population. If not, please address this selection bias in the discussion.

6 – Line 166-167. "Middle Eastern", while an important subcategory, is not a part of the NIH definition of underrepresented groups. If you would like to use the NIH guidelines, please remove this subcategory from the UR coding. Alternatively, please directly justify the coding of "Middle Eastern" as underrepresented. Please see revisions #13 and #14 for further discussion about NIH guidelines.

7 – Please ensure that the in-text reported percentages and N values are easy to understand, as they are quite unclear in the current version. This is particularly the case in the "participants" section of the methods and figure legends, where it is unclear what year each number represents.

Methodological and statistical issues:

8 – The data collected in 2019 and 2020 was combined for data analysis, suggesting that some participants may have participated in both years. Logistic regression requires the observations to be independent of each other: that is, the observations should not come from repeated measurements. This could bias/inflate the results – the larger number of repeated respondents, the higher chance to obtain biased results – and it therefore needs to be carefully justified and addressed. If you can identify repeated respondents, please report how many, and justify why you think you can include them in the analysis: in addition, please consider exploring whether your findings stay the same if you take out those repeated respondents, and/or use a repeated measures design to compare the difference in mental health between 2019 and 2020. If you are not able to identify repeated respondents, please discuss this limitation.

9 – The mental health outcomes are sensitive to recent events and affected by many factors/variables (e.g., variables related to the pandemic, years in the degree, disciplines, etc.): it is possible that the mental health disparities you report might be no longer statistically significant once these covariates are included in the analysis. Please either include these covariates in your logistic regression, or explain in the manuscript why they were not included.

10 – The collected mental health measures could be treated as continuous variables, which could offer better variabilities for statistical modelling. Please clarify why you decided to simplify the scale (0 and 1) to run logistic regression analyses (binary outcome) rather than multiple regressions (continuous outcome). If you are not able to provide the reason, please make sure that both approaches (logistic regression and multiple regression) yield consistent findings.

Additional analyses and discussion points pertaining to intersectionality approaches:

11 – Please consider conducting additional analyses to explore whether they were any differences across gender identity. The N values seem similar to the other populations you have chosen to compare, and this would significantly increase the impact of this study. Otherwise, please justify why this variable was not explored.

12 – If feasible, please adopt an intersectionality approach, in particular by examining how gender AND race/ethnicity impact your findings. Otherwise, please justify why these analyses were not performed, and strengthen your discussion around intersectionality.

13 – While Asians are included as well-represented according to NIH standards, the percentage of Asian-identifying individuals in the sample (at 16% and 9%) clearly demonstrates that they are not. This is discussed at the end, but please also include these comments in the Results section and strengthen the language. In addition, please consider comparing the experiences of Asian and White students, or, if you are concerned with anonymity/statistical power, please consider comparing WR vs. UR where "Asian" is moved from WR to UR, as this would allow the reader to understand if coding "Asian" as WR vs. UR makes a difference (this does not necessarily need to be in the main part of the paper). Alternatively, please justify why these analyses were not performed.

14 – On that note: although discussed at the end, please provide a more nuanced discussion of race and ethnicity, and strengthen your stance. In particular, while it is understandable that you want to use NIH standards, please acknowledge more overtly that race is a social construct (so as to not reinforce the notion that these categories may be biological); and please discuss in more depth how the NIH categories may be lacking and not best represent a marginalized identity in science.

Strengthening certain claims

15 – Please provide evidence for the following statements:

a – Line 40 ("biomedical graduate programs are exposed to unique stressors by the nature of their training"): please consider moving up certain citations (e.g. Hazell et al., 2020 and Mackie and Bates 2019) to support this affirmation. Please also consider replacing the word "unique" as long hours, pressure to produce, and unsupportive mentors may not necessarily support a "uniquely" stressful situation.

b – Line 93 ("Trainees of color, especially people identifying as Black, Indigenous, and people of color (BIPOC), may also be affected by heightened concerns about individual, community, and family health."): While covid disparities are certainly real, and likely at play, please provide supporting evidence for this affirmation.

c – Line 340 ("We also speculate that being in a medical training program with obvious direct impact during a time when medical professionals were publicly celebrated had a positive effect on MD student mental health.") This discussion of MDs being "publicly celebrated" and "heroic" and able to connect their training to the crisis compared to PhD lacks evidence, particularly as we see protests outside of hospitals continue.

d – Line 343 ("Furthermore, medical students were able to continue some form of their training remotely, maintain some social contact and peer support networks through academic training, and avoid graduation delays.") Please provide citations for this statement, and clarify whether this is institution-specific.

e – Line 356 ("Biomedical PhD student training was severely curtailed in the 2020 spring months…"). Please clarify if this refers to specific measures at your institution.

f – Line 410 ("the mental health crisis in graduate biomedical education has only recently received considerable attention from the laboratory research community."): Please provide evidence for this statement.

---

## [Author Response]

Essential revisions:Comparing medical and PhD students:1 – Please consider discussing in the introduction (and the discussion) why there may be a difference between medical and biomedical PhD students. In particular, please expand on:a. The level of institutional support provided by the respective programs: For instance, what are the mental health resources provided to medical and PhD students? Are they the same and are they easily accessible e.g. paid for? facilitated/triaged? on-campus?b. The awareness of the two populations of their mental health (one might argue that medical students are trained to recognize certain signs and symptoms).c. The differences that are intrinsic to their training: for example, those familiar with the medical student training process might suggest more acute stressors but more structured support due to LCME requirements.d. The response to the COVID pandemic from the two different programs: for instance, the curriculum and/or level of support changed for the medical students in 2020, which is highlighted in the discussion. However, this is a big limitation in interpreting the data (comparatively between MD and PhD students). This can be viewed as an intervention that the PhD students did not have.

We thank the reviewers for this important point and have added the following to the discussion to address these valid critiques:

“In contrast, PhD students continued work at best, or experienced forced absence from the lab, with no specific intervention addressing emerging COVID knowledge or mental health coping strategies.”

“Furthermore, accreditation requirements for MD training provide a more structured support system for medical student mental health and wellness compared with PhD student supports. Given the difference in support offered due to enforced standards, this makes MD support services more easily accessible and freely available (no cost, and/or available funding to cover any out-of-pocket mental health medical expenses). Whereas both MD and PhD have availability of limited initial contact of facilitated and triaged on-campus resources, MDs have access to 1.5 FTE dedicated support for 800 students (e.g., 533:1 student-counselor ratio) whereas PhDs have only.5 FTE dedicated support for 600 students (e.g., 1200:1 student-counselor ratio). The combination of the more robust support for MD students with the removal of exposure to acute stressors often encountered in clinical rotations may have magnified this effect.”

2 – Following that thought: medical and graduate students have extremely different training paradigms and, in this institution, medical students received an intervention that could have mitigated the effects of the pandemic on their mental health. Therefore, please consider whether directly comparing medical to graduate students is relevant, and if so, how to do so: for example, the bar graphs side by side are fine, but the chi-sq tests may be an unfair comparison. Or at least, please discuss the caveat of the different training paradigms and levels of support provided throughout the 2020 year.

We agree that the nuanced discussion of differences between the two populations is very important, and brings to attention the point that adjustment of the training environment (e.g., MD intervention) can in fact positively impact mental health even amidst difficult times. We have replaced the post hoc chi-squared tests with a more complex logistic regression model that allows for statistical control of differences between the two populations, including marginal means and conditional odds ratios to expresses in terms of differences in each population, while simultaneously controlling for the other variables’ impact.

3 – The reason why PhD students have worse PHQ9 outcomes over time comparatively is because the medical students show a decrease from 2019 to 2020. Could you please report whether you have looked at the trends across the medical student years (1 through 4), and justify if not.

We agree that analysis by year in program would be important to include in future work; however, in our sample this was a demographic question that was not completed by a sizeable portion of respondents, which would reduce our ability to complete the main analysis if we included it. We also have concerns about including variables with a significant amount of missing data as this would potentially not be representative of the sample or population.

Discussing the sample, and how it was coded:4 – Line 158: Please detail the criteria for sample invitations (e.g. are there any inclusion or exclusion criteria for the study, and if so, are they the same for the 2019 and 2020 surveys?).

The same listservs were used each year to recruit School of Medicine trainees in Medical Education (MD students) and Graduate Education (biomedical PhDs), as described in Methods. A standard IRB-approved invitation included a brief paragraph explaining the survey, signed by the appropriate administrator (e.g., SOM Wellness Coach). There were no formal inclusion or exclusion criteria in either year other than being enrolled in either the MD or PhD programs, and anyone included on the SOM trainee listservs was invited to respond.

5 – Line 158: Please clarify whether the respondents (n=431, Fall 2020; n=526, Fall 2019) are a representative sample that closely matches the characteristics of your population. If not, please address this selection bias in the discussion.

Unfortunately, we can’t directly assess how well our samples were representative of their respective populations, the following has been added in limitations to address this:

“Due to anonymous data collection and optional questions to protect participant identities, we cannot assess granular response rates by demographic characteristics. Future studies could be completed with a controlled sample matched with participant identifiers to know who in the sample is and isn’t responding, as this could impact findings.”

6 – Line 166-167. "Middle Eastern", while an important subcategory, is not a part of the NIH definition of underrepresented groups. If you would like to use the NIH guidelines, please remove this subcategory from the UR coding. Alternatively, please directly justify the coding of "Middle Eastern" as underrepresented. Please see revisions #13 and #14 for further discussion about NIH guidelines.

Thank you for pointing out the limitations of NIH definition, here and elsewhere – we have further explained the reasoning for inclusion of Middle Eastern and expanded the discussion of why this category was included. In addition, we adjusted the terminology to better reflect the categories included in our analyses to include identities that have been historically excluded. This category has been retitled accordingly to, “historically excluded (HE)” as reflected in science by race/ethnicity and gender (HE gender discussed elsewhere). Please also see other author responses for a summary; as well as expanded responses about marginalized identity, gender, intersectionality, Asian identity, and an expanded discussion of NIH definition along with limitations thereof.

The following has been added:

“Further, the stress experienced by Asian Americans as scapegoats of the pandemic and targets of anti-Asian violence may also affect the mental health of Asian American students within academia.” (Introduction)

“In the primary analysis, mental health data was analyzed by type of training program (MD vs. PhD), year (2019 vs. 2020), and historically excluded (HE) populations by race/ethnicity (HE-RE vs. NHE-RE) and by gender (HE-G women vs. NHE-G men). A post hoc analysis included the primary variables as wells as historically excluded by sexual orientation (HE-SO LGBQ+ vs. NHE-SO non-LGBQ+).” (Methods)

“in accordance with National Institutes of Health definitions of underrepresented in the biomedical, clinical, behavioral, and social sciences (NIH, 2020). Historically, Asian Americans have also faced exclusion from American culture as evidenced by Japanese Internment and the Chinese Exclusion Act. However, Asian Americans have been well-represented in the sciences and thus are included as NHE-RE for analysis. Additionally, Middle Eastern is not included in the US Census or NIH definitions as underrepresented; however, this identity group is included as HE-RE due to the marginalization especially over the past two decades in the United States following 9/11/2001.” (Methods-Participants)

7 – Please ensure that the in-text reported percentages and N values are easy to understand, as they are quite unclear in the current version. This is particularly the case in the "participants" section of the methods and figure legends, where it is unclear what year each number represents.

We have simplified the figures and legends in favor of an expanded unified description in the Methods, as we recognize that the prior version may be confusing. Previously, the chi-squared post hoc analysis relied on specific n and degrees of freedom for each test that impacted interpretation (no longer crucial for the current analyses). Especially given the updated modeling with interaction terms used to replace the previous chi-squared post hoc tests, the exact n per each subgroup is less important and instead the odds ratios and p-values are the primary focus. Accordingly, to reflect this change, Figure 1 has been simplified per the editors and reviewers’ suggestions; Figure 2 and Figure 3 (new Figure 3 added to visualize effects identified in the post hoc analysis) also now include percentages embedded in the figures rather than embedding these details in the legends.

Methodological and statistical issues:8 – The data collected in 2019 and 2020 was combined for data analysis, suggesting that some participants may have participated in both years. Logistic regression requires the observations to be independent of each other: that is, the observations should not come from repeated measurements. This could bias/inflate the results – the larger number of repeated respondents, the higher chance to obtain biased results – and it therefore needs to be carefully justified and addressed. If you can identify repeated respondents, please report how many, and justify why you think you can include them in the analysis: in addition, please consider exploring whether your findings stay the same if you take out those repeated respondents, and/or use a repeated measures design to compare the difference in mental health between 2019 and 2020. If you are not able to identify repeated respondents, please discuss this limitation.

Due to data being collected anonymously, we cannot confirm if there were any repeated participants across the year timepoints; we recognize that this is a limitation, and have added the following to the discussion to acknowledge this:

“We recognize that if a large number of participants responded in both samples this could be problematic for assumptions of independence for use of parametric statistics and the Logistic Regression analysis used specifically, hence results should be interpreted with caution. Because we collected data anonymously to protect respondents’ privacy, we cannot assess to what extent there may have been respondents who participated in both years. If we did have identifiable data and were able to use a repeated-measures design, this would be preferable to reduce error variance. To the extent that significant findings were achieved even with the increased error variance inherent in a between-subjects design, these results likely had large enough effect sizes to be identified even given the loss of power from using a between-subjects design rather than the more sensitive within-subjects design that would be preferable.”

9 – The mental health outcomes are sensitive to recent events and affected by many factors/variables (e.g., variables related to the pandemic, years in the degree, disciplines, etc.): it is possible that the mental health disparities you report might be no longer statistically significant once these covariates are included in the analysis. Please either include these covariates in your logistic regression, or explain in the manuscript why they were not included.

We did not collect pandemic-specific data as we used the same annual survey questions from the previous year, future evaluations should ask and control for these variables. Limitations for being able to include years in training is discussed elsewhere (see additional author responses to year in training variable). We did not collet specialty or departmental affiliations to protect anonymity, to increase comfort with responding, and to maximize response rates. Furthermore, disciplinary clusters were deemed homogeneous enough to be considered in the two major categories of biomedical PhD training and medical MD training.

“Additional limitations included our lack of ability to control for other possible variables of interest such as pandemic-specific factors, years in training, and departmental affiliation or specialty area. Pandemic-specific questions were not asked because we used the same annual survey questions from the previous year to maintain comparable responses. Department and specialty information were not asked in order to protect anonymity, to increase comfort with responding, and to maximize response rates. Yet, populations were purposefully defined by reasonably homogenous training experiences into the two major clusters of interest: biomedical PhD training and MD training. Optional demographic questions were not completed for number of years in training for a large portion of the sample, limiting our ability to include this in the analysis. Future work should consider controlling for as many of these variables as possible.”

10 – The collected mental health measures could be treated as continuous variables, which could offer better variabilities for statistical modelling. Please clarify why you decided to simplify the scale (0 and 1) to run logistic regression analyses (binary outcome) rather than multiple regressions (continuous outcome). If you are not able to provide the reason, please make sure that both approaches (logistic regression and multiple regression) yield consistent findings.

The research question of interest was best captured by a bivariate designation of symptomatic versus asymptomatic (or problematic use versus non-problematic use). Hence, we approached the question from a bivariate logistic regression approach to answer the question accordingly. We felt this was the most appropriate model to answer the research question, and could be best interpreted to determined when respondents were functionally affected by their mental health; hence our decision to define it as such in the statistical modeling. Severity of symptoms at a more granular level, especially relying on self-report approaches rather than clinical diagnoses, posed several limitations that we felt made it a less appropriate approach to the data. Furthermore, for implications and policy purposes, we feel that the defined research question best illustrates the sweeping problem of mental health in graduate biomedical and medical education.

Additional analyses and discussion points pertaining to intersectionality approaches:11 – Please consider conducting additional analyses to explore whether they were any differences across gender identity. The N values seem similar to the other populations you have chosen to compare, and this would significantly increase the impact of this study. Otherwise, please justify why this variable was not explored.

We agree that including gender as variable would add to the intersectional applicability of our model, and hence have added it as a control variable for all primary analyses.

We also decided that if evaluating the impact of gender were to be added to the model, we would be remiss to not also include sexual orientation status if we had a large enough sample size, as the intersectionality of gender and sexual identity can interact with race/ethnicity, trainee type, and year as well. However, we were also cautious about inadvertently reducing power to evaluate main effects beyond the primary research questions in order to ensure enough power to evaluate the primary intersectional identities (race/ethnicity, gender, and trainee type) by year. Hence, we added a post hoc analysis that accounted for both gender and LGBQ status using an intersectional approach.

We have added sections in the results and discussion describing the revised primary analysis (adding gender using a more sophisticated Logistic Regression model including interaction terms, conditional odds ratios, and marginal means to replace the more simplistic Chi-Squared comparisons previously displayed) as well as the post hoc model which includes LGBQ identity. Along with these substantive changes to the analyses, we have included justification and precedent for including both gender and sexual orientation based on previous studies. Transgender identity could not be included due to the small sample size, but should be included in future work and this limitation is also addressed in the manuscript.

Relatedly, to avoid reductions in power we also ran post hoc tests to further examine the impact of Asian social identities in a more granular way, and have reported the new results accordingly (using the model that also controls for gender).

12 – If feasible, please adopt an intersectionality approach, in particular by examining how gender AND race/ethnicity impact your findings. Otherwise, please justify why these analyses were not performed, and strengthen your discussion around intersectionality.

We have adopted an intersectional approach, with additional citations to justify this in the introduction as well as a deeper dive in the discussion. Thank you for this suggestion, we believe it has strengthened our approach and the applicability of the findings greatly.

13 – While Asians are included as well-represented according to NIH standards, the percentage of Asian-identifying individuals in the sample (at 16% and 9%) clearly demonstrates that they are not. This is discussed at the end, but please also include these comments in the Results section and strengthen the language. In addition, please consider comparing the experiences of Asian and White students, or, if you are concerned with anonymity/statistical power, please consider comparing WR vs. UR where "Asian" is moved from WR to UR, as this would allow the reader to understand if coding "Asian" as WR vs. UR makes a difference (this does not necessarily need to be in the main part of the paper). Alternatively, please justify why these analyses were not performed.

Thank you for this suggestion. We agree that a more granular investigation of the impacts on Asian participants is useful, despite having a small sample once dividing HE-RE into 3 classifications; hence have included a supplemental post hoc analysis as mentioned in other author responses which still allows us to address the questions while retaining power for the primary analysis. The following has been added:

“People who identified of Asian descent generally scored better than other NHE or HE groups, indicating fewer mental health challenges on all four outcome variables and the three suicidal ideation variables. However, conditional odds rations based on these models indicated some changes by year that may be associated with the negative psychosocial and environmental factors many Asian Americans experienced between 2019-2020. Accounting for the effect of race (3-level) while controlling for year, MD/PhD, and gender in 2020 compared with 2019, Asian participants showed greater mean depression scores (p<.03) and anxiety scores (p<.04); there was no change in either alcohol use (p=.07, though marginal) or drug use (p=.28). There were no significant changes in conditional probabilities for suicidal ideation of Asian participants during that time. Future directions should include examination of the intersectionality of these and other identity groups, and systemic barriers that each may encounter differentially. Follow-up studies should explore the impact of racism on Asian Americans and contrast Asian American experiences with White American experiences when reporting HE-RE data.”

14 – On that note: although discussed at the end, please provide a more nuanced discussion of race and ethnicity, and strengthen your stance. In particular, while it is understandable that you want to use NIH standards, please acknowledge more overtly that race is a social construct (so as to not reinforce the notion that these categories may be biological); and please discuss in more depth how the NIH categories may be lacking and not best represent a marginalized identity in science.

Thank you for these valuable suggestions, which we have leaned on heavily to reframe much of our discussion and additional analyses. We believe this has strengthened our findings and our ability to interpret implications more broadly. We have: (a) expanded our discussion of intersectionality; (b) added more nuanced analysis controlling for a wider range of identities (including the addition of gender to all of our primary models; c) included a post hoc analysis and discussion of Asian identity as well as sexual orientation); (d) re-envisioned the terminology used to Historically Excluded (HE) in science (HE-RE and HE-G); (e) we have expanded our discussion of limitations to the NIH definitions of WR/UR; and f) we have explicitly noted the social derivation of race.

The following has been added to the discussion (see NIH definition expanded discussion in other author responses):

“We recognize [both that race is a social construct], and that other social identity groups beyond race and ethnicity (Rothman, Gunturu, Korenis, 2020) may also experience inequitable impacts of COVID-19 and similar structural biases may negatively impact mental health for other reasons (Hunt, 2020). (Discussion)

“In fact, the NIH has recently expanded its definition of underrepresented in science to include women, people with disabilities, first generation college students, and those from disadvantaged backgrounds, including people who have experienced or are experiencing homelessness, foster care participation, recipients of free and reduced lunch, Pell grants, SNAP or WIC, and those who grew up in a low-income or rural areas (NIH, 2020; NIH, 2018, rescinded and replaced in 2019; NIH, 2019). This expanded definition is still flawed as some groups are still excluded from recognition and inclusion as societal norms and practices shift over time. For instance, people of middle eastern descent who have faced heightened discrimination, especially following 9/11/2001.” (Discussion)

NIH definition limitations are discussed in a number of other points in the response as well (please see also other author responses regardig Middle Eastern and Asian identities, gender, intersectionality, and NIH definition limitations).

Furthermore, particularly in recent American history, it is noteworthy that over the past 20 years many groups have been systematically marginalized during different periods of time. Future definitions of historically excluded groups may also want to consider the impact of current events with politics and their intersection with discrimination against specific social identities. We must acknowledge that science is not immune from such effects, which can drive out a talented, diverse pool of future scientists in training.

Strengthening certain claims15 – Please provide evidence for the following statements:a – Line 40 ("biomedical graduate programs are exposed to unique stressors by the nature of their training"): please consider moving up certain citations (e.g. Hazell et al., 2020 and Mackie and Bates 2019) to support this affirmation. Please also consider replacing the word "unique" as long hours, pressure to produce, and unsupportive mentors may not necessarily support a "uniquely" stressful situation.b – Line 93 ("Trainees of color, especially people identifying as Black, Indigenous, and people of color (BIPOC), may also be affected by heightened concerns about individual, community, and family health."): While covid disparities are certainly real, and likely at play, please provide supporting evidence for this affirmation.c – Line 340 ("We also speculate that being in a medical training program with obvious direct impact during a time when medical professionals were publicly celebrated had a positive effect on MD student mental health.") This discussion of MDs being "publicly celebrated" and "heroic" and able to connect their training to the crisis compared to PhD lacks evidence, particularly as we see protests outside of hospitals continue.d – Line 343 ("Furthermore, medical students were able to continue some form of their training remotely, maintain some social contact and peer support networks through academic training, and avoid graduation delays.") Please provide citations for this statement, and clarify whether this is institution-specific.e – Line 356 ("Biomedical PhD student training was severely curtailed in the 2020 spring months…"). Please clarify if this refers to specific measures at your institution.f – Line 410 ("the mental health crisis in graduate biomedical education has only recently received considerable attention from the laboratory research community."): Please provide evidence for this statement.

Unique has been replaced as follows:

“Considering that students in medical school and in biomedical graduate programs are exposed to a multitude of stressors by the nature of their training including long hours, pressure to produce, and unsupportive mentors…”

B – We have reframed the terminology to center on Historically Excluded instead of BIPOC; nonetheless, we have also included an additional reference to support the extension of the COVID-19 impacts to disproportionately impact BIPOC individuals, communities, and family health.

We have also added the following reference, examining HE-RE working class inequities due to COVID-19:

Pathak, E. B., Menard, J. M., Garcia, R. B., and Salemi, J. L. (2021). Social Class, Race/Ethnicity, and COVID-19 Mortality Among Working Age Adults in the United States. medRxiv.

Feldman, J. M., and Bassett, M. T. (2021). Variation in COVID-19 Mortality in the US by Race and Ethnicity and Educational Attainment. JAMA network open, 4(11), e2135967-e2135967.

Limas, J.C. (2021). Adaptation to Overexpression of Cyclin E in Epithelial Cells. (Publication No. 2616907910) [Doctoral dissertation, University of North Carolina at Chapel Hill]. ProQuest Dissertations and Theses Global.

Limas, J. C., Corcoran, L. C., Baker, A. N., Cartaya, A. E., and Ayres, Z. J. (2022). The Impact of Research Culture on Mental Health and Diversity in STEM. Chemistry–A European Journal, e202102957.

“While our students themselves may be classified as associated with some lower risk groups for direct effects COVID due to their levels of education, BIPOC students disproportionately hail from working-class families and identify as first-generation college graduates. Hence, students historically excluded by race and ethnicity (HE-RE; e.g., those identifying as BIPOC) may experience disproportionate impact to their families, in that their families have been most at risk for mortality from COVID during 2020 within each social class (Pathak et al., 2021). Similarly, highest risk of age-adjusted mortality during COVID was identified for Hawaiian and Other Pacific Islander, American Indian or Alaska Native, and Latinx or Hispanic people (Black, Feldman and Bassett, 2021). An illustrative example includes a PhD sharing instances of her own experience as a Mexican-American woman PhD trainee during this time (Limas, 2021).”

C – We acknowledge the very real impact of anti-medical establishment influences as well, and have added the following to address this oversight: “Even so, this was not a ubiquitous experience, as simultaneously continued protests and divisiveness regarding vaccination were also pervasive across the country, which could ameliorate this effect (e.g., Bogel-Burroughs, 2020).”

Bogel-Burroughs, N. (2020). Antivaccination activists are growing force at virus protests. The New York Times. May, 2.

D and E – We have clarified these statements, and qualified them by specifying, “at our institution.”

F – The following has been added to clarify:

“Whereas medical schools have had mental health recommendations in place since 1992 (AAMC, 1992; enforced by Liaison Committee on Medical Education accreditation standards), biomedical PhD programs have had no such parallel policies regarding mental health provisions nationwide. Similarly, while there have been frequent calls for mental health research in higher education and there is already a robust body of literature decades old already in medical education (e.g., Slavin, 2016; a systematic review of medical education mental health evidence-based research included 195 relevant studies, Rotenstein et al., 2016) , evidence-based research on mental health research of PhDs has begun to emerge over the past five years (e.g., Levecque et al., 2017; Evans et al., 2018). Resources have been mobilized to evaluate and respond to the needs of medical students (e.g., 28 studies evaluating interventions, Wasson et al., 2016), which indeed still merit additional attention given high rates of depression, anxiety, and suicidality.”

The following additional references have also been added:

AAMC Executive Council (1992). Recommendations Regarding Health Services for Medical Students. Washington, DC: Association of American Medical Colleges

Available: https://www.aamc.org/professional-development/affinity-groups/gsa/health-services-recommendations

Wasson, L. T., Cusmano, A., Meli, L., Louh, I., Falzon, L., Hampsey, M., … and Davidson, K. W. (2016). Association between learning environment interventions and medical student well-being: a systematic review. Jama, 316(21), 2237-2252.